# A method for semantic textual similarity on long texts

Omar Zatarain[1], Juan Carlos González-Castolo[2] and
Silvia Ramos-Cabral[1]

[1] Department of Computer Science and Engineering, University of Guadalajara, Ameca, Jalisco,
Mexico
[2] Department of Information Systems, University of Guadalajara, Zapopan, Jalisco, Mexico

## ABSTRACT

This work introduces a method for the semantic similarity of long documents using sentence transformers and large language models. The method detects relevant information from a pair of long texts by exploiting sentence transformers and large language models. The degree of similarity is obtained with an analytical fuzzy strategy that enables selective iterative retrieval under noisy conditions. The method discards the least similar pairs of sentences and selects the most similar. The preprocessing consists of splitting texts into sentences. The analytical strategy classifies pairs of texts by a degree of similarity without prior training on a dataset of long documents. Instead, it uses pre-trained models with any token capacity, a set of fuzzy parameters is tuned based on a few assessment iterations, and the parameters are updated based on criteria to detect four classes of similarity: identical, same topic, concept related, and non-related. This method can be employed in both small sentence transformers and large language models to detect similarity between pairs of documents of random sizes and avoid truncation of texts by testing pairs of sentences. A dataset of long texts in English from Wikipedia and other public sources, jointly with its gold standard, is provided and reviewed to test the method's performance. The method's performance is tested with small-token-size sentence transformers, large language models (LLMs), and text pairs split into sentences. Results prove that smaller sentence transformers are reliable for obtaining the similarity on long texts and indicate this method is an economical alternative to the increasing need for larger language models to find the degree of similarity between two long texts and extract the relevant information. Code and datasets are available at: https://github.com/omarzatarain/long-texts-similarity. Results of the adjustment of parameters can be found at https://doi.org/10.6084/m9.figshare.29082791.

Corresponding author
Omar Zatarain,
omar.zatarain@academicos.udg.mx



## INTRODUCTION

Semantic textual similarity is critical for achieving machine understanding in computational linguistics and artificial intelligence. Textual similarity has been studied at the sentence level with good results (*Tian et al., 2017*; *Shao, 2017*; *Sultan, Bethard & Sumner, 2015*; *Reimers & Gurevych, 2019*). Despite the promising results for semantic textual similarity of snippets, research on long texts is scarce. The emergence of attention

models (*Vaswani et al., 2017*; *Zhang et al., 2020*; *Sukhbaatar et al., 2019*; *Shaw, Uszkoreit & Vaswani, 2018*; *Roy et al., 2021*; *Zamani et al., 2018*; *Choromanski et al., 2020*) increased the number of tokens that can be processed on several tasks, including summarization Longformer (*Beltagy, Peters & Cohan, 2020*), Bigbird (*Zaheer et al., 2020*), BART (Bidirectional and Auto-Regressive Transformers) (*Lewis et al., 2020*), GPT 2 (*Radford et al., 2018*), Unlimiformer (*Bertsch et al., 2023*) and natural language inference. From these large language models, only the Unlimiformer (*Bertsch et al., 2023*) has no strict limit on the number of tokens. The attention of these models is implemented through expensive matrix operations and queries and requires big amounts of memory and processing time. To reduce the complexity in space and time several attention strategies have been proposed (*Baevski & Auli, 2019*; *Sukhbaatar et al., 2019*; *Shaw, Uszkoreit & Vaswani, 2018*; *Roy et al., 2021*; *Zamani et al., 2018*; *Choromanski et al., 2020*; *Hofstätter et al., 2020*; *Press, Smith & Lewis, 2021*; *Dao et al., 2022*; *Bertsch et al., 2023*). A model devoted to semantic textual similarity on long texts (*He et al., 2024*) tailors the texts up to 1,024 tokens and uses a dataset of Chinese. In addition, there is a lack of a long text dataset and a gold standard for comparing pairs of texts. Fuzzy logic enables reasoning under uncertain conditions (*Zadeh, 1965*, *1999*) and establishes a framework to reason according to a scale of similarity values. The systematic comparison of embeddings from a pair produces, in most cases, low similarity and a small fraction of pairs have high similarity.

## RELATED WORK

This section explores the state of the art in three main dimensions related to the task of long-text similarity: (1) models, (2) datasets for training the models, and (3) metrics for assessing the performance of the models on the pursued tasks.

### Models

Table 1 describes the state of the art of language models, their objectives, types of attention, employed metrics, and the capacity size in tokens. Research on text similarity on short texts has been extensive, thanks to a contest on semantic text similarity (*Cer et al., 2017*; *Tian et al., 2017*; *Shao, 2017*; *Sultan, Bethard & Sumner, 2015*), the most accurate methods for short text similarity use embeddings (*Pennington, Socher & Manning, 2014*; *Mikolov & Zweig, 2012*) and exploit attention with transformers (*Vaswani et al., 2017*). Attention strategies include self-attention (*Vaswani et al., 2017*), sliding window (*Baevski & Auli, 2019*); adaptive span (*Sukhbaatar et al., 2019*); sparse attention (*Shaw, Uszkoreit & Vaswani, 2018*; *Roy et al., 2021*; *Zamani et al., 2018*; *Choromanski et al., 2020*), local self-attention (*Hofstätter et al., 2020*), flash attention (*Dao et al., 2022*), short attention (*Press, Smith & Lewis, 2021*), attention by chunks (*Bertsch et al., 2023*). These techniques aim to extract relevant information and, at the same time, be efficient in the processing. A model for semantic similarity on text snippets, *Reimers & Gurevych (2019)* uses an architecture with a classification objective function and a regression objective for training and cosine similarity for similarity testing. Research on transformers for long-text similarity is scarce; the task with the most research on long texts is summarization. The

**Table 1 Summary of models compared in SOTA for text similarity and other related tasks.**

| Model | Objectives | Attention | Metrics | Size type |
|---|---|---|---|---|
| Sentence Bert (*Reimers & Gurevych, 2019*) | Semantic text similarity | Classification objective function, regression objective function, triplet objective function | Pearson | Short texts |
| LongFormer (*Beltagy, Peters & Cohan, 2020*) | Summarization | Dilated sliding window + global attention with projections | ROUGE-1, ROUGE-2, ROUGE-L | Large (up to 16,384 tokens) |
| BigBird (*Zaheer et al., 2020*) | Summarization, Genomics | Generalized attention mechanism | ROUGE, bits per character | >3,000 tokens |
| BART (*Lewis et al., 2020*) | Summarization, Generation tasks, Natural language understanding (NLU), Stanford sentiment treebank (SST), Semantic text similarity, Natural Language Inference (NLI) | Full attention (bidirectional) with left to right decoder | Accuracy, F1-score, perplexity, ROUGE-L, BLEU | Large (8,000 tokens) |
| GPT 2 (*Radford et al., 2018*) | NLI, question answering (QA), Sentence similarity, classification | Multi-headed self attention | Accuracy, F1, Matthews correlation, Pearson correlation | Medium (512 tokens) |
| Unlimiformer (*Bertsch et al., 2023*) | Summarization | BERT Attention and k-nearest neighbors (KNN) search | ROUGE, BERTScore, Entity Mention Recall | Unlimited |
| Match unity (*He et al., 2024*) | Semantic text similarity | Global (Longformer) and sliding window | Accuracy, F1-score | Large texts up to 1024 tokens |

semantic text similarity of long texts is scarce; only two works address the topic, *Jiang et al. (2019)* and *He et al. (2024)*. However, several models from other tasks are explored on long texts: Longformer (*Beltagy, Peters & Cohan, 2020*), Bigbird (*Zaheer et al., 2020*), BART (*Lewis et al., 2020*), GPT 2 (*Radford et al., 2018*), Unlimiformer (*Bertsch et al., 2023*). A study on the self-attention of Bidirectional Encoder Representations from Transformers (BERT) models (*Kovaleva et al., 2019*) reveals that the position of tokens affects the performance of the model; there is no strong relation between the weights of tokens and their linguistic semantics parts-of-speech (POS). A study of the performance of language models focused on long contexts (*Liu et al., 2024*) and applied to multi-document question-answering provides evidence that the positions of words can decrease the model's performance. This phenomenon is described as a U-shape performance curve (*Laming, 2010*). The study included an algorithm called FlashAttention, which uses sequence lengths up to one hundred thousand tokens (*Dao et al., 2022*). Besides the models developed for text snippets, only the Match Unity model (*He et al., 2024*) is devoted to long-text similarity up to 1,024 tokens. Research on the effectiveness of long-context transformers (*Qin, Feng & Van Durme, 2023*) shows evidence on the limited capacity of transformers to increase the accuracy on long texts. Before the boom of language models, natural language processing was focused on the structure of concepts and their relations, as example, *Mooney & DeJong (1985)* describes the semantics into schemes that consider the part of speech and the structure. A study on the effects of using POS in text similarity (*Zatarain et al., 2023*) provides evidence of the positive impact of considering grammar despite the absence of training in the system.

## Datasets

Current datasets for semantic text similarity (*Cer et al., 2017*; *Wang et al., 2018*; *Dolan & Brockett, 2005*) are designed on text snippets (sequences smaller than 1,024 tokens). The datasets implemented on semantic text similarity are focused mainly on tasks of summarization and some in text similarity: standardized comparison over long language sequences (SCROLLS) (*Shaham et al., 2022*), Text REtrieval Conference (TREC) (*Craswell et al., 2021*), Semantic Text Similarity Benchmark (STS-B) (*Cer et al., 2017*), General Language Understanding Evaluation (GLUE) (*Wang et al., 2018*), Microsoft Research Paraphrase Corpus (MSRP) (*Dolan & Brockett, 2005*), WikiText-103 (*Merity et al., 2017*), Quora (*Chen et al., 2018*), PG-19 (*Sun et al., 2021*). SCROLLS (*Shaham et al., 2022*) is a dataset for long text sequences and is compiled for summarizing documents, natural language inference, and other tasks in a standardized form on documents with sizes up to $10^5$ tokens. However, this dataset has not been tested for text similarity. The TREC (*Craswell et al., 2021*) deep learning track for document retrieval and passage retrieval and uses four classes of relevance for both tasks and is based on the Microsoft Machine Reading Comprehension (MS-MARCO) dataset (*Bajaj et al., 2018*) consisting of 3.5 million web documents among other resources. STS-B (*Cer et al., 2017*) is a dataset for comparing pairs sentences with a man-made gold standard. GLUE (*Wang et al., 2018*) contains STS-B among other datasets; however, it does not provide a dataset for semantic text similarity on large texts. MSRP (*Dolan & Brockett, 2005*) is a large dataset of sentence pairs distilled from the internet, using the Levenshtein distance. WikiText-103 (*Merity et al., 2017*) is a Wikipedia articles dataset. Quora (*Chen et al., 2018*) is a question-pairs dataset for determining if the questions are duplicated; the questions are curated to be concise. PG-19 (*Sun et al., 2021*) is a book dataset for summarization with books with publication dates before 1919.

## Metrics

The main metrics used in sentence similarity are the Pearson correlation, the Spearman correlation and for the case of long text similarity, accuracy and F1-score (*Powers, 2020*; *He et al., 2024*). All metrics use the similarity degree in the real interval [0, 1]; the gold standard is described as a set of real values from a human perspective. Related tasks such as summarization or question answering use Recall-Oriented Understudy for Gisting Evaluation (ROUGE) (*Lin, 2004*) or Bilingual Evaluation Understudy (BLEU) metrics (*Papineni et al., 2002*). In the semantic similarity of long texts, Pearson and Spearman correlations can be applied at the sentence level; however, additional criteria should be used due to the presence of pairs with multiple sentences instead of single-sentence pairs.

## CONTRIBUTION

This work proposes a method for semantic text similarity on long texts that uses analysis by parts and exploits language models of any size. The method avoids biases due to the maximal token capacity of language models and reduces the complexity in terms of memory and specialized hardware such as graphic processing units (GPUs). A dataset of

long texts is compiled, and experiments on pairs of texts from the dataset use sentence models and large language models to assess the similarity.

# MATHEMATICAL MODELS

The proposed method is based on a set of analytic equations that extract the degrees of similarity. One key aspect of the systematic similarity comparison of sentences or texts is the noise or low degree of similarity of present unrelated texts, including identical texts. Therefore, the decision on the similarity of a pair of texts is made on a small number of pairs of sentences. In this sense, the method applies selective attention by discarding the noisy pairs of sentences.

**Definition 4.1** (Set of comparisons (SC)). Let $S_1$ and $S_2$ the sets of sentences within $Text_1$ and $Text_2$, respectively. The universe set of comparisons $SC$ consists of the pairs of sentences defined by the Cartesian product of sentences from $S_1$ and $S_2$, *i.e.*,

$$SC = S_1 \times S_2. \tag{1}$$

**Definition 4.2** (Noise). The *Noise* $\subset SC$ is the set of comparisons with a low degree of similarity (Sim) than a threshold $v$[1]:

$$Noise(SC) = \bigcup_{i=1}^{n} (s_{1i}, s_{2i}) \in SC | Sim(s_{1i}, s_{2i}) \leq v. \tag{2}$$

The noise set represents the sentences that will be disregarded in the text's quantitative and qualitative analysis. The noise parameter $v$ is set based on comparing distributions of pairs deemed as not similar. The noise distribution is particular to each model due to the training performed and the dataset used for the purpose.

**Definition 4.3** (Spanning). The spanning is the highest decile with non-zero pairs of sentences.

$$Spanning = max\_level(nonzero(deciles)). \tag{3}$$

The spanning provides the criterion of confidence of two text comparisons and boosts the elicitation of the highest similarities of pairs of sentences.

**Definition 4.4** (Support). Let Deciles be the intervals (10% cohorts) for the distribution of comparisons of sentence pairs in SC (1). The support is the minimum decile, where more relevant pairs of sentences are accumulated from the maximum non-empty decile to achieve the $M$ most relevant pairs of sentences from the highest decile to the lower decile. The parameter $M$ is the number of sentences of the smallest document.

$$Support = arg\_min(deciles), \left( \sum_{i=spanning}^{support} deciles(i) \right) \geq M. \tag{4}$$

The support represents the level of similarity that accumulates at least M pairs of sentences with the higher similarities starting from the spanning downwards. The support aims to set the boundary between the relevant pairs of sentences and the noise to facilitate attention.

[1] The estimation of $v$ is achieved by weighing the highest Decile that concentrates low similarities in pairs of the four classes. The estimation depends on the distribution of pairs of sentences deemed as non-related in a set of pairs of documents.

**Table 2 Example of a long text pair similarity comparison by getting the distributions per deciles of their sentence pairs.**

| | Distribution per deciles (number of sentence pairs) | | | | | | | | | |
|---|---|---|---|---|---|---|---|---|---|---|
| Comparison | D1 | D2 | D3 | D4 | D5 | D6 | D7 | D8 | D9 | D10 |
| Same topic | 22,195 | 35,159 | 35,144 | 34,559 | 14,619 | 3,039 | 383 | 46 | 8 | 0 |

**Definition 4.5 (Soundness).** The soundness is the decile with the highest number of relevant pairs of sentences that exists between the support and the spanning.

$$Soundness = max(deciles), support \leq soundness \leq spanning. \tag{5}$$

**Definition 4.6 (Classification of texts according to similarities).** Let x be the average value of similarity of sentence pairs from the soundness Eq. (5) to the spanning, four fuzzy sets of similarity are defined regarding the pairs of text: non-related (NR), concept-related (CR), same topic (ST), and identical (I).

$$NR(x, \alpha) = 1 - \left( \frac{1}{1 + e^{-\alpha \times 10(x \times 10 - \alpha \times 10)}} \right) \tag{6a}$$

$$CR(x, b, \beta) = e^{-\frac{(x-\beta)^2}{2b^2}} \tag{6b}$$

$$ST(x, c, \gamma) = e^{-\frac{(x-\gamma)^2}{2c^2}} \tag{6c}$$

$$I(x, \delta) = \frac{1}{1 + e^{-\delta \times 10(x \times 10 - \delta \times 10)}}. \tag{6d}$$

In Eq. (6a) regarding to the assessment of non-related documents, $\alpha$ represents the decreasing starting point of the inverse sigmoidal for non-related documents. Equation (6b), corresponding to documents that share some concepts, considers the variables b and $\beta$ as the width and the center of the Gaussian membership of the concept-related documents. Equation (6c) assesses documents describing the same topic, which has the variables c and $\gamma$ the width and the center of the Gaussian membership of the same topic ST. Finally, Eq. (6d), which identifies documents containing identical contents (plagiarism), uses the variable $\delta$ as the width of the increasing ramp and the starting of the constant maximum value of the sigmoidal distribution. As an example, consider the comparison of two related texts in Table 2; the texts have been formatted in the sets of sentences A = 274 sentences, B = 497 sentences, the set of comparisons Eq. (1) SC = $A \times B$ = 145, 152 sentence pairs. From SC, sentences belonging to lower deciles are disregarded since they lack relevant semantics Eq. (2) $N(SC)$ = 127, 057 sentence pairs, with $v$ = 0.4 in a conservative estimation[2]. The spanning Eq. (3) provides the reference to the highest non-zero decile for applying the selection; in this case, spanning = 9th decile. The support Eq. (4) with $M = argmin (A, B)$ = 274 sentences is the 7th decile. The soundness Eq. (5) is also the 7th decile.

**Definition 4.7 (Assessment of the degree of similarity of a pair of documents).** The classification of a pair of texts $Class(T_1, T_2)$ is, therefore, determined by the maximum degree of membership from the sets NR, CR, ST and I.

$$Class(T_1, T_2) = argmax(NR, CR, ST, I). \tag{7}$$

[2] The estimation of $v$ in this example is set due to the observations of the distributions of pairs of documents as described forward in 'Dataset of random-size texts' and Fig. 3.

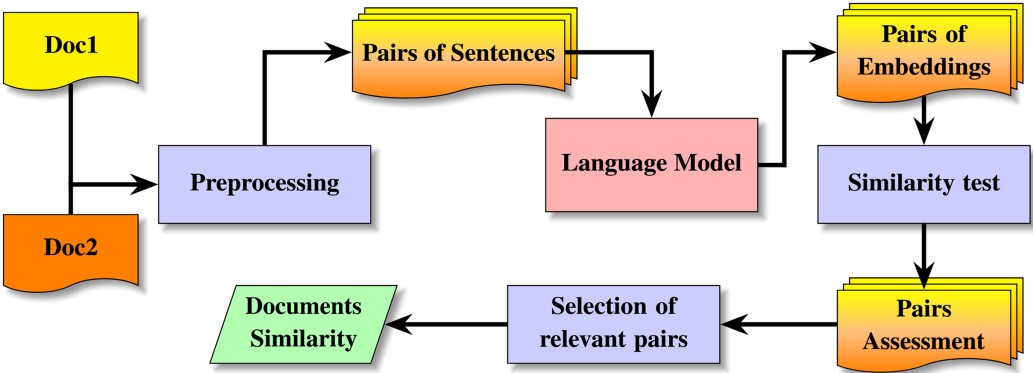

**Figure 1 Method for detection of semantic text similarity on texts of random-size.**

---

**Algorithm 1  Attention on long texts.**

**Require:** $LT1, LT2$: Pair of long texts; *Model*: The used model; $[\alpha], [b, \beta], [c, \gamma], [\delta]$:
parameters for non-related, concept-related, same topic and identical fuzzy sets

**Ensure:** *Class_data*: structure of the classification of both texts; *SentPairs*: the indexes of
the most similar sentences; *Data*: structure of the distribution and analysis of similarity

1: $Sent1 \Leftarrow SplitintoSentences(LT1)$

2: $Sent2 \Leftarrow SplitintoSentences(LT2)$

3: $Emb1 \Leftarrow GetEmbeddings(Sent1)$

4: $Emb2 \Leftarrow GetEmbeddings(Sent2)$

5: $PairData.Deciles \Leftarrow Array[10]$

6: $PairData.Matrix \Leftarrow Matrix[\|Sent1\|][\|Sent2\|]$

7: **for** $e_1 \in Emb1$ **do**

8:      **if** $validSentence(Sent1[pos(e_1)])$ **then**

9:          **for** $e_2 \in Emb2$ **do**

10:             **if** $validSentence(Sent2[pos(e_2)])$ **do**

11:                 $similarity \Leftarrow cosine(e_1, e_2)$

12:                 $PairData.Matrix[pos(e_1)][pos(e_2)] \Leftarrow similarity$

13:                 $index \Leftarrow GetDecile(similarity)$

14:                 $increase(Data.Deciles[index])$

15:             **end if**

16:         **end for**

17:     **end if**

18: **end for**

19: $PairData.Support \Leftarrow GetSupport(PairData.Deciles)$

20: $PairData.Spanning \Leftarrow GetSpanning(PairData.Deciles)$

21: $PairData.Soundness \Leftarrow GetSoundness(PairData.Deciles)$

22: $Class\_data \Leftarrow ClassifyPair(PairData, a, b, c, d, \alpha, \beta, \gamma, \delta)$

---

*(Continued)*

| Algorithm 1 (continued) |
| --- |
| 23: $SentPairs \Leftarrow SelectRepresentativePairs(PairData)$ |
| 24: $saveResults(Pairdata, SentPairs, Class\_data)$ |

## METHOD FOR DETECTION OF SIMILARITY ON LONG TEXTS

Figure 1 describes the method for acquiring the similarity between texts of random size using low-size or large-size language models. The method starts the preprocessing, which splits each document into sentences. For each sentence in each document, the model produces the corresponding embedding. From the sentence embeddings, a systematic comparison of pairs of embeddings returns a distribution by deciles; from the pairs of documents, the method produces the spanning Eq. (3), the support Eq. (4) and the soundness Eq. (5). Algorithm 1 describes the details of this process for selective attention using a model and extracting the semantic text similarity from two random-size texts and splitting texts into sentences. The algorithm for splitting by chunks is similar to Algorithm 1, the difference between both versions of the algorithm is the use of the function *SplitintoChunks* instead of *SplitintoSentences*. The process starts the decomposition of both texts into sentences, or into chunks[3] of fixed-size instead of sentence, however, this option is not recommended due to potential biases[4]. For each sentence, the model provides the embedding. The array of deciles is initialized for recording the distribution of similarities. The attention requires the systematic cosine similarity between each tuple $(e_1, e_2) \in Emb_1 \times Emb_2$, where $Emb_1$ and $Emb_2$ are the sets of sentence embeddings from the pairs of texts. The testing of attention occurs only if both sentences are validated[5], this validation mitigates biases due to truncated texts due to models capacity. It follows the analysis of the spanning Eq. (3), the support Eq. (4), and the soundness Eq. (5). The assessment of the pair *Class_data* Eq. (7) is achieved by selecting the most accurate membership at the fuzzy sets Eqs. (6a)–(6d). Finally, the indices of the most relevant pairs of sentences are selected from the matrix of similarities *PairData.Matrix*.

### Complexity of the method and processes

The attention complexity of a hypothetical systematic comparison of a pair of documents is $O (N \times M)$, where $N$ is the number of sentences of the first document, and $M$ is the number of sentences of the second document. The complexity of the attention is $O(N \times M)$, where $N$ and $M$ stand for the number of sentences in each text instead of the number of tokens.

## DATASET OF RANDOM-SIZE TEXTS

After a bibliographical review, we found no previous specification of a dataset of random-size English texts devoted to semantic similarity. The dataset consists of 72 documents with random sizes of tokens. The sizes of the documents in the dataset vary from a few hundred to twenty-four thousand words, as described in Fig. 2. The number of sentences varies from tens to below 1,500 sentences. The dataset produces 2,627 combinations of document pairs for testing the method. The distributions for the

[3] Only one option is allowed when comparing a pair of texts, splitting by sentences or splitting by chunks.

[4] The text can be decomposed into chunks of fixed size instead of decomposing by sentences, however, it was found that chunking produces more biases regardless of the used model; for this reason, it is not recommended to chunk texts with the proposed method.

[5] It was encountered during the experiments that several false positives are due to noun phrases present in non-related documents, *e.g.*, "The U.S." phrase appears in several non-related documents at the dataset in this work.

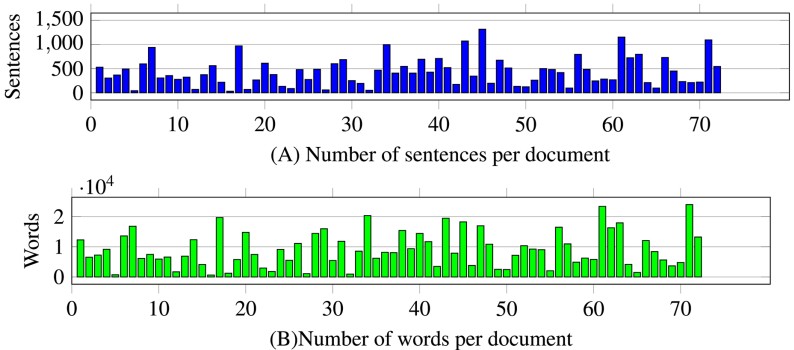

**Figure 2 Number of sentences and words by document of the dataset.** The number of documents in the dataset is 72. The maximum number of sentences in documents is above 1,000, and the maximum number of words is above 24,000.

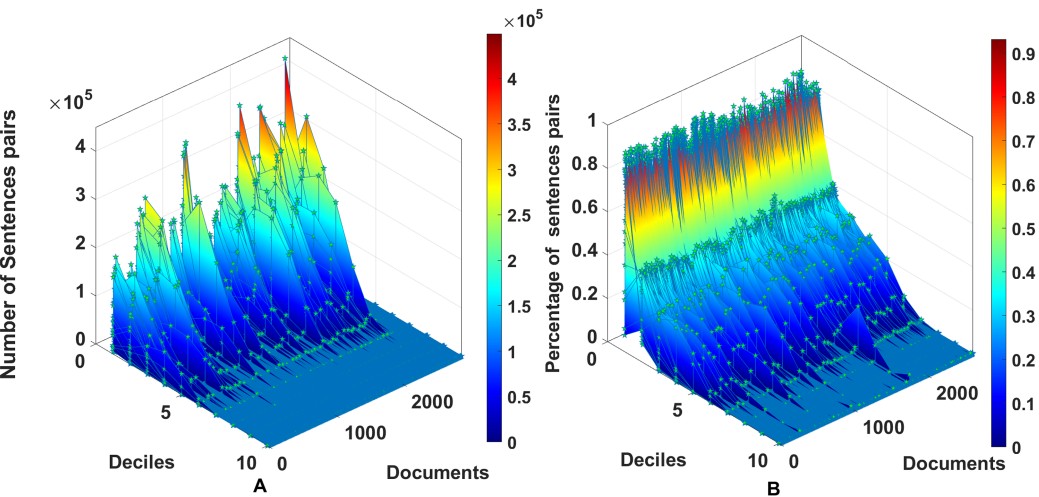

**Figure 3 Distribution of pairs of documents within the dataset of 72 documents, using the model all-MiniLM-L6-v2.** The number of text pairs is 2,627, including the 72 *self-comparisons* for each document.

systematic 2,627 dataset comparisons from 72 documents in Fig. 3 show high concentrations in the lower deciles starting from fifth decile, to prevent biases with pairs having a degree of similarity, the threshold $v$ is set to 0.4[6].

## Gold standard of the dataset

A gold standard is produced for the dataset of 72 long texts; each pair of documents is labeled empirically by commonsense. At the initial labeling using the commonsense; most text pairs are non-related, 72 pairs are deemed identical, 29 pairs are considered the same topic, 193 pairs are concept-related, and the remaining 2,333 are deemed as non-related. The gold standard was created by common sense on the documents' titles and later validated with the method's results; the validation enabled the adjustment of parameters in equations to detect the similarity degree of each document Eqs. (6a)–(6d).

[6] The estimation of $v$ varies with the distribution of similarity produced by each model; the distribution depends on the criteria for training the models, having in mind that non-related pairs of texts produce non-zero degree of similarity from the embeddings.

**Table 3 Sentence transformers and large language models tested with the proposed method.**

| Model | Model specification | | | | | |
|---|---|---|---|---|---|---|
| | Param. (millions) | Memory usage (GB) | Hyper-param token train | Max input tokens | Emb dim | GPU |
| all-MiniLM-L6-v2[1] | 22.7 | 0.09 | 128 | 256 | 384 | No |
| all-MiniLM-L12-v2[2] | 33.4 | 0.12 | 128 | 256 | 384 | No |
| all-mpnet-base-v2[3] | 110 | 0.41 | 128 | 384 | 768 | No |
| glove.6B.300d[4] | 120 | 0.45 | – | – | 300 | No |
| Longformer[5] | 102 | 0.59 | – | 16,000 | 768 | Yes |
| BigBird[6] | 1 | 16.0 | 512 | 4,096 | 768 | Yes |
| GPT-2[7] | 117 B | 0.55 | – | 1,024 | 768 | Yes |
| BART[8] | 139M | 0.55 | – | 1,024 | 4,096 | Yes |

**Notes:**
[1] https://huggingface.co/sentence-transformers/all-MiniLM-L6-v2.
[2] https://huggingface.co/sentence-transformers/all-MiniLM-L12-v2.
[3] https://huggingface.co/sentence-transformers/all-mpnet-base-v2.
[4] https://huggingface.co/sentence-transformers/average_word_embeddings_glove.6B.300d.
[5] https://github.com/allenai/longformer
[6] https://huggingface.co/docs/transformers/model_doc/big_bird.
[7] https://huggingface.co/docs/transformers/model_doc/gpt2.
[8] https://huggingface.co/docs/transformers/model_doc/bart.

### Tuning of assessment fuzzy sets

Once the gold standard for the dataset is created, a review process tunes the assessment fuzzy parameters of Eqs. (6a)–(6d). Assessing the performance of the language model with the proposed method and setting the parameters requires experimentation on a dataset labeled by humans. Labels on pairs of documents are standardized into four types of similarity: (1) identical (I) Eq. (6d), (2) same topic (ST) Eq. (6c), (3) concept related (CR) Eq. (6b) and (4) non-related (NR) Eq. (6a).

## EXPERIMENTS

### Experiment 1: generation of similarity comparisons

The experiment applies the method to each pair of long texts from Wikipedia and other sources; each text is converted into sentences, and the embeddings of sentences are obtained with one of the models described in Table 3. After obtaining the similarity Soundness Eq. (5), the label of the pair is predicted given a configuration of parameters and compared against the gold standard of the pair. The number of pairs of sentences is determined by $N \times M$, where N is the number of sentences in document A and M is the number of sentences in document B. The gold standard is defined in terms of four discrete labels: identical, same topic, concept-related, and non-related. The assessment of parameter configurations from Eqs. (6a)–(6d) is applied to four sentence transformers, BART, Longformer, BigBird, and GPT2.

### Experiment 2: assessment of results and tuning of parameters

The assessment of results is statistically performed through the true positives (TP), false positives (FP), and false negatives (FN) described in *Powers (2020)*; the true negatives (TN)
[7] The experiment considers a multi-class context; for this reason, the accuracy does not include true negatives to avoid making the results on the classes with small numbers of members vanish.

are not used due to the experiment has a multiclass context[7]. The performance of the method is measured with the metrics of $precision = \frac{TP}{TP+FP}$, $recall = \frac{TP}{TP+FN}$, $accuracy = \frac{TP}{TP+FP+FN}$, and $F_1 = 2 \times \frac{precision \times recall}{precision+recall}$. These equations are applied for each class of similarity degree (I, ST, CR, and NR). An iterative process produces the tuning for the parameters of Eqs. (6a)–(6d). The process is applied to each model described in Table 3. Initially, the parameters are set to any values within the range [0, 1], with the unique condition being $v \leq \alpha < \beta < \gamma < \delta$. Based on the assessment forward in 'Results of Tuning of Parameters and Assessment', the parameters are updated until no improvement in results.

## Hardware/software setups

One hardware setup is used for experimentation on eight models and five software configurations. Larger models require more hyperparameters and computational power involving GPUs. The setup runs in an affordable computation scenario (PC or laptop). The smaller models (sentence transformers and Global Vectors (GloVe)) can run with or without GPUs; large language models require at least a GPU as a prerequisite.

**Hardware Setup.** The hardware setup involves an Intel i7 processor, 12 GB RAM, and a GPU model Nvidia Geforce GTX 1050.

**Software Setups.** The software setups are A Python 3.10.11 using the transformers package, PyTorch, for the large language models: BirBird, Longformer, GPT2, and BART, where each model is implemented separately in a customized configuration. For the smaller models (the sentence transformers and Glove), the configuration includes Python 3.10 with the package of sentence transformers.

## RESULTS

### Results of comparison of similarity for each model

The results of the similarity extraction are measured with a subset of 28 document pairs, the pairs are representative due to the size (some of the largest with variety on the four similarity classes). Figure 4 shows the soundness Eq. (5) achieved by each of the eight models. The sentence models are more sensitive than GloVe and the large language models. The implementation of the cosine similarity of sentence models and the large language model is different; sentence models produce sentence embeddings with a standardized size, whereas large language models provide embeddings at the level of tokens, which requires that the sets of embeddings for both sentences be adjusted to the same size, either through tailoring the longer set of embeddings or filling the shorter set of embeddings; the strategy was filling the shorter one with underscore characters. The results of large language models show that only identical and non-identical pairs of sentences can be detected, and the explicit classes of similarity (ST, CR, and NR) are, in most cases, indistinguishable. On the other hand, the sentence models exhibit greater sensitivity and produce a more varied level of soundness for each pair in the benchmark, which provides room for establishing the similarity classes.

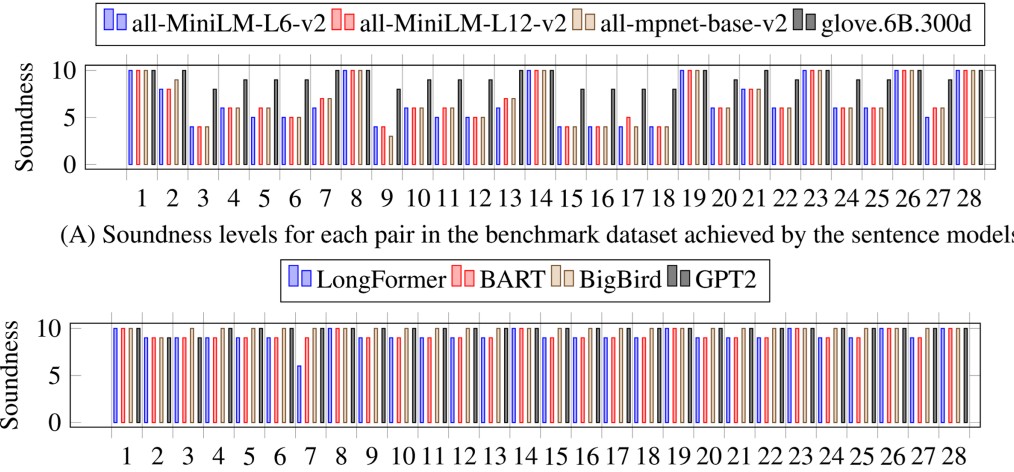

**Figure 4 Levels of achieved soundness with the benchmark dataset of 28 pairs and comparing the eight language models.** (A) Sentence models, (B) Large language models. All models are tested with preprocessing using sentences. Sentence models are more sensitive to differences than large language models.

**Table 4 Configurations for assessing the models' performance.**

| | | Assessment classes | | | | | |
| | | NR | CR | | ST | | I |
| Configuration | $v$ | $\alpha$ | b | $\beta$ | c | $\gamma$ | $\delta$ |
|---|---|---|---|---|---|---|---|
| Baseline 0 | 0.3 | 0.3 | 0.2 | 0.5 | 0.2 | 0.7 | 0.9 |
| Baseline 01 | 0.4 | 0.5 | 0.2 | 0.6 | 0.2 | 0.7 | 0.9 |
| Baseline 02 | 0.5 | 0.6 | 0.2 | 0.64 | 0.2 | 0.7 | 0.9 |
| MiniL12 | 0.5 | 0.55 | 0.2 | 0.65 | 0.2 | 0.75 | 0.9 |
| MPNET 1 | 0.5 | 0.55 | 0.2 | 0.67 | 0.2 | 0.75 | 0.9 |
| Glove 1 | 0.6 | 0.7 | 0.2 | 0.8 | 0.2 | 0.9 | 0.95 |
| Glove 2 | 0.6 | 0.7 | 0.2 | 0.8 | 0.2 | 0.9 | 0.95 |
| Longformer 1 | 0.8 | 0.9 | 0.2 | 0.94 | 0.2 | 0.95 | 0.98 |
| Longformer 2 | 0.8 | 0.88 | 0.2 | 0.92 | 0.2 | 0.94 | 0.995 |
| BART 1 | 0.8 | 0.86 | 0.2 | 0.88 | 0.2 | 0.90 | 0.98 |

## Results of tuning of parameters and assessment

The set of tested configurations on the sentence and large language models is enumerated in Table 4. Table 5 contains the results of testing configurations with the language model all-MiniLM-L6-v2. The first configuration Baseline 0 has the minimum level of soundness of three, the maximum level of soundness is 10, therefore the level of noise $v$ is set as 0.3, consequently the conservative estimation of the parameter $\alpha = 0.3$ has the purpose of weighing possible relevant pairs of sentences to prevent biases. Configuration Baseline 0 is applied to the dataset, which produces the non-related pairs being detected as concept-related. Therefore, the Baseline 01 configuration increases noise $v$, $\alpha$, and $\beta$.

**Table 5 Assessment of configurations using the model all-MiniLM-L6-v2.**

| Configuration | | Confusion matrix | | | | Performance | | | |
|---|---|---|---|---|---|---|---|---|---|
| Dataset | Class | I | ST | CR | NR | Prec. | Rec | F1 | Acc. |
| | I | 72 | 1 | 0 | 0 | 1.0 | 0.98 | 0.99 | 0.98 |
| Baseline 0 | ST | 0 | 28 | 68 | 13 | 0.96 | 0.2 | 0.4 | 0.25 |
| Dataset 72 | CR | 0 | 0 | 125 | 2,190 | 0.64 | 0.05 | 0.09 | 0.05 |
| | NR | 0 | 0 | 0 | 130 | 0.05 | 1.0 | 0.1 | 0.05 |
| | I | 72 | 1 | 0 | 0 | 1.0 | 0.98 | 0.99 | 0.98 |
| Baseline 01 | ST | 0 | 28 | 68 | 13 | 0.96 | 0.25 | 0.40 | 0.25 |
| Dataset 72 | CR | 0 | 0 | 122 | 211 | 0.63 | 0.36 | 0.46 | 0.30 |
| | NR | 0 | 0 | 3 | 2,109 | 0.90 | 0.99 | 0.94 | 0.90 |
| | I | 72 | 1 | 0 | 0 | 1.0 | 0.98 | 0.99 | 0.98 |
| Baseline 02 | ST | 0 | 28 | 68 | 13 | 0.96 | 0.25 | 0.40 | 0.25 |
| Dataset 72 | CR | 0 | 0 | 0 | 0 | 0.0 | 0.0 | 0.0 | 0.0 |
| | NR | 0 | 0 | 125 | 2,320 | 0.99 | 0.94 | 0.97 | 0.94 |

**Table 6 History of results on testing the models using the dataset of 72 documents and 2,628 pairs (998 in the case of LongFormer, BigBird, BART and GPT2). The metrics values are the average of the four classes.**

| Model | Configuration | Precision | Recall | Accuracy | F1 |
|---|---|---|---|---|---|
| all-MiniLM-L6-v2 | Baseline 0 | 0.7375 | 0.5425 | 0.59 | 0.5425 |
| all-MiniLM-L6-v2 | Baseline 01 | 0.8725 | 0.645 | 0.6975 | 0.6075 |
| all-MiniLM-L12-v2 | Baseline 02 | 0.775 | 0.565 | 0.5825 | 0.505 |
| all-MiniLM-L12-v2 | MiniL12_2 | 0.8475 | 0.825 | 0.8225 | 0.7325 |
| all-mpnet-v2 | MPNET 1 | 0.8575 | 0.7925 | 0.8075 | 0.715 |
| glove.6B.300d | Baseline 01 | 0.3 | 0.0955 | 0.1385 | 0.0956 |
| glove.6B.300d | Glove 1 | 0.365 | 0.347 | 0.14825 | 0.10 |
| glove.6B.300d | Glove 2 | 0.5525 | 0.36975 | 0.3475 | 0.24425 |
| Longformer | Longformer 1 | 0.485 | 0.2874 | 0.33 | 0.2775 |
| Longformer | Longformer 2 | 0.49 | 0.3125 | 0.195 | 0.1125 |
| BigBird | Longformer 1 | 0.2537 | 0.025 | 0.0155 | 0.00775 |
| BART | BART 1 | 0.3055 | 0.23175 | 0.022 | 0.113 |
| GPT-2 | Longformer 1 | 0.254 | 0.2365 | 0.016 | 0.008 |

Baseline 01 produces still non-related pairs which are detected as concept related; Baseline 02 is defined with an increasing value of $\alpha$ and $\beta$. Baseline 02 produces no improvement; therefore, the configuration tuning process stops. Based on the best results provided by the configuration at Table 5, Baseline 01 applied on the model all-MiniLM-L6-v2, the performance on the F1 metric is low for the pairs of documents in the classes ST and CR; these gaps indicate that the model has a coarse sensitivity when minor differences exist between the assessment parameters. This phenomenon is replicated with various degrees in the rest of the tested models and configurations as described in Table 6. Potential causes are the lack of models' training and/or fine-tuning, considering the method's classes.

Another cause is the internal strategy to distribute the similarity degrees, for example, consider the found levels of noise throughout the tested configurations in Table 4; the level of noise directly affects the performance, the more evenly distributed pairs, the more capacity to deduce the degree of similarity.

### Dataset labeling validation

The validation of the dataset consists of a systematic verification of the assessment with the parameters and comparison against the gold standard. If the predicted class mismatches the label considered at the gold standard, the pair of documents is reviewed to see if it was wrongly labeled; if the label at the gold standard is correct, then the parameters are updated accordingly. The review is facilitated with a software tool that retrieves pairs of sentences from documents which disagree on the labeling.

## Ablation study

An ablation study with sentence and large language models is applied to observe differences in the distribution and performance by testing several configurations. In the case of the large language models, and due to the large amount of time required to produce the analysis of a pair, only the first 998 pairs are analyzed. In the case of sentence-transformers, the dataset is analyzed in full. Table 6 contains the configurations tested with the dataset of 72 documents. For the sentence language models, the model all-mpnet-v2 with the Masked and Permuted Pre-trained Network (MPNET) 1 configuration achieves the best results. However, the model all-MiniLM-L6-v2 achieves the highest precision with the Baseline 01 configuration. The large language models achieved low performance due to their soundness in the interval [0.9, 1.0].

Figure 5 shows the distributions per deciles without noise removal and the three configurations (Baseline 0, Baseline 01, and Baseline 02), and the model all-MiniLM-L6-v2. The distributions are normalized for simplicity. In (A), the noise is not removed; more similar pairs of sentences vanish due to the vast amount of low-similarity pairs. (B) depicts the noise removal with $v = 0.3$; in this case, more similarity pairs are enacted. (C) increases the noise removal to $v = 0.4$; in this case, more relevant pairs are enacted in comparison to (B). (D) shows the removal to $v = 0.5$; this removal produces null similarity in all deciles for three cases, this removal produces biases; therefore, (C) is the appropriate noise removal for the model all-MiniLM-L6-v2.

## Memory and time required by the models

This section examines the time and memory requirements for determining the similarity of text pairs. The comparison of the text pairs from scratch using the hardware setup and software setups at 'Hardware/Software Setups' includes the eight models' implementations of the method with the benchmark of 28 pairs from seven documents extracted from the dataset of 72 documents. The criteria for choosing the subset of documents are: documents with larger sizes, some documents with a high degree of similarity, including non-related documents. Table 7 shows the time and memory used by each implementation; the

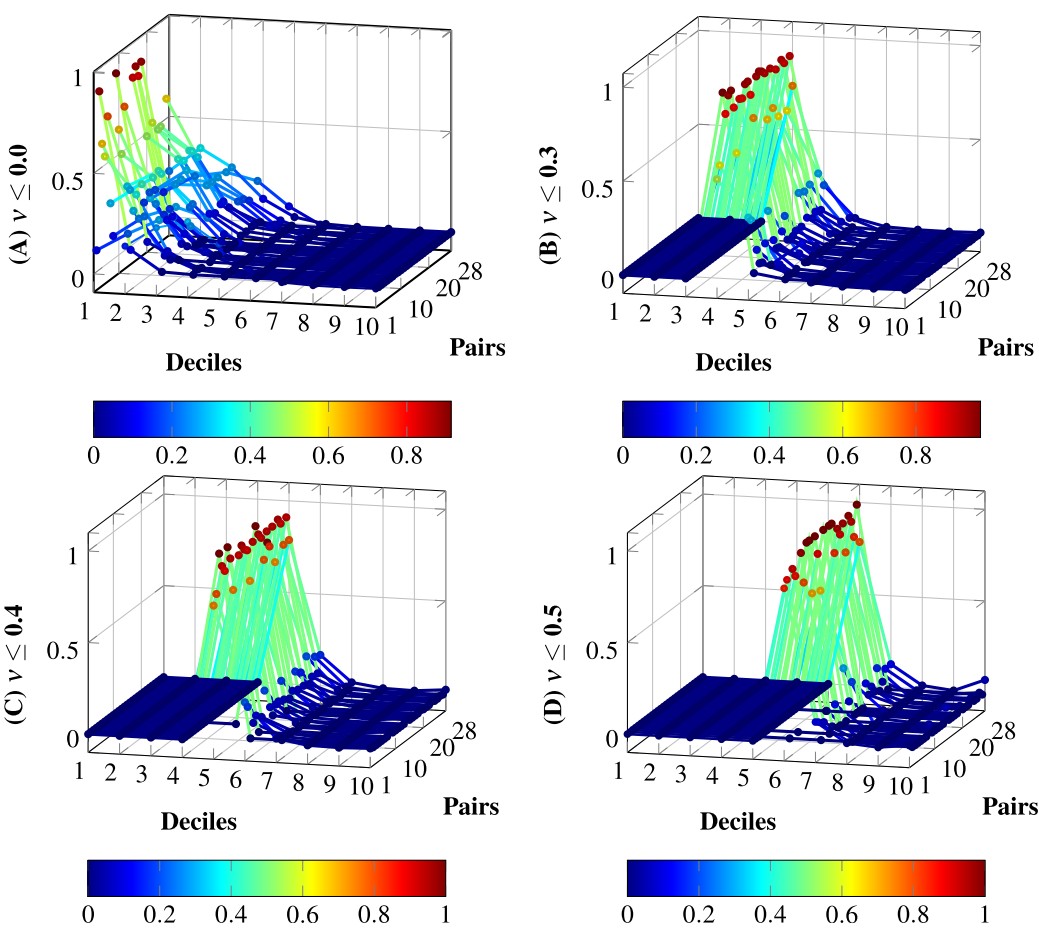

**Figure 5 Distributions with and without noise thresholds $v$ applied to the model all-MiniLM-L6-v2 on the dataset of 28 pairs.** (A) Normalized distribution without tailoring: $v = 0.0$. (B) Normalized distribution with noise threshold $v = 0.3$, from configuration Baseline 0. (C) Normalized distribution with noise threshold $v = 0.4$, from configuration Baseline 01. (D) Normalized distribution with noise threshold $v = 0.5$, from configuration Baseline 02. The tailoring in (D) produces total biases in some distributions; the tailoring in (C) does not produce total biases and increases the degree of membership for certain deciles in comparison to (B). Therefore, the appropriate level of noise for the model with the dataset is set to $v = 0.4$.

sentence models require less time and memory, whereas the large models require more memory and use more time; the overhead due to the use of the GPU increases the computing time and the model size demands more memory resources. The minimum average time for obtaining the similarity results is achieved by the GloVe model, at 78 s, and the maximum average time is 1,828 s (30.4 min). The rest of the sentence models have an average performance of less than 3.5 min, whereas the minimum average of large language models is BigBird with 11.23 min. The minimum time for obtaining a pair was 13 s from the GloVe model, compared with 3.9 min for the GPT2 model; the difference is 17 times faster for GloVe than GPT2. The most time-efficient model is GloVe, with a time of 127 s, and the least time-efficient model is LongFormer. The use of a GPU does not

**Table 7 Results on time and memory usage models with a benchmark of seven documents and 28 pairs.**

| Model | Total (sec) | Avg (sec) | Min(sec) | Max (sec) | Memory (MB) |
|---|---|---|---|---|---|
| all-MiniLM-L6-v2 | 2,936 | 105 | 22 | 163 | <2,000 |
| all-MiniLM-L12-v2 | 3,549 | 127 | 30 | 190 | <1,600 |
| all-mpnet-base-v2 | 5,088 | 182 | 49 | 262 | <1,600 |
| glove.6B.300d | 2,180 | 78 | 13 | 127 | <2,500 |
| LongFormer | 50,165 | 1,792 | 231 | 3,347 | <5,500 |
| BART | 49,252 | 1,759 | 331 | 2,974 | <6,200 |
| BigBird | 18,859 | 674 | 85 | 1,187 | <6,300 |
| GPT-2 | 51,190 | 1,828 | 238 | 3,109 | <5,700 |

improve because the size of sentence pairs from documents is small, and the overhead when a GPU becomes expensive.

## DISCUSSION

This method works despite not being trained with the specific topics and the sets of similar pairs because it uses two forms of cognition. The first is recognition using a language model and exploiting its pretraining. The second is reasoning under uncertain conditions with fuzzy inferences on the soundness Eq. (5). According to the experiment results, large language models require additional fine-tuning, since they were not specifically trained for this task. Further research on exploiting large language models would lead to efficient architectures that may reduce processing time by embedding the proposed method for parallelizing the processing of sentence pairs. As shown in experiments, this method can be used with any pre-trained model with general or specific datasets and requires no additional information to detect the similarities between texts. The accuracy of the implementation on a model depends on the pre-training and the selection of the assessment parameters (values of the Eqs. (6a)–(6d), (7)). Any model is handled as a black box; therefore, this method is suitable to be implemented in parallel architectures for time efficiency.

## LIMITATIONS

The primary disadvantage of this method is that the tuning of the assessment parameters depends on the efficiency of training a language model, and the biases inherent in the latter affect the method's performance, including partial or total false positives and false negatives. The biases can be due to scarce information at training and the pursued objective. Additionally, there are defined degrees of similarity, rather than predefined classes based on specific content. Another disadvantage of the method with large language models (LLMs) is that the capacity of the latter is sub-used, preventing the high efficiency that may be achieved by exploiting the full resources of an LLM;

nonetheless, this opens new opportunity areas to boost the processing time for long text similarity.

## CONCLUSION

The proposed method for measuring long-text similarity is optimized using sentence transformers for selective attention to the most similar sentences within a pair of long texts. This selective attention mechanism allows for the retrieval of the most similar parts of documents while disregarding pairs of less similar sentences. The process for generating gold-standard data facilitates the analysis of pairs of long texts, enabling a comprehensive assessment of the method's implementation across various models. The practical applications of this method include text retrieval from long documents and text similarity analysis of long texts, without requiring prior data or training on the entire content of the documents.

## ACKNOWLEDGEMENTS

We thank all the reviewers for their comments; their comments helped improve this article.

This research exploits the sentence language models of all-MiniLM-L6-v2, all-MiniLM-L12-v2, all-mpnet-v2, glove.6B.300d and the large language models of Longformer, BigBird, BART and GPT2 to obtain the embeddings from the sentences of documents; the embeddings of sentences are used to compute the degree of similarity and produce the similarity of pairs of texts through of the proposed method.

### Funding
The authors received no funding for this work.

### Competing Interests
The authors declare that they have no competing interests.

### Author Contributions
- Omar Zatarain conceived and designed the experiments, performed the experiments, performed the computation work, prepared figures and/or tables, authored or reviewed drafts of the article, and approved the final draft.
- Juan Carlos González-Castolo analyzed the data, authored or reviewed drafts of the article, dataset curation, gold-standard definition, and approved the final draft.
- Silvia Ramos-Cabral analyzed the data, authored or reviewed drafts of the article, dataset curation, gold-standard definition, and approved the final draft.

## Data Availability

The dataset used for experiments, input files, code, gold standard, and results are available at GitHub:

- https://github.com/omarzatarain/long-texts-similarity. Follow the instructions in the Quick_Start.md file to reproduce the experiments of the method with the eight models.

Also available at Zenodo:

omarzatarain. (2025). omarzatarain/long-texts-similarity: long-texts-similarity (v1.0.0). Zenodo. https://doi.org/10.5281/zenodo.16889826.

## Supplemental Information

Supplemental information for this article can be found online at http://dx.doi.org/10.7717/peerj-cs.3202#supplemental-information.

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
