# Peer review of "A method for semantic textual similarity on long texts"

_PeerJ Computer Science, doi:10.7717/peerj-cs.3202_

## Round 0.1 · original submission · Major Revisions

Both reviewers proposed very valuable comments to improve the manuscript. All comments must be addressed by the author.

·

Basic reporting

The paper is well-structured and demonstrates a reasonably good level of English. However, several points require improvement to enhance clarity and readability. Below are the detailed suggestions:
In Related Work
The related work section could be better organized. The authors are encouraged to introduce subsections such as:
-Models
-Datasets
-Metric
A summary table is recommended to improve clarity. The table could outline the models, their objectives, types of attention mechanisms used, and evaluation metrics.
At line 122, the authors reintroduce the concepts of transformers and attention, which were already mentioned earlier when discussing various models. This repetition affects the document's readability. It would be better to introduce these concepts earlier or use a more appropriate transition at line 122.
Similarly, the readability of lines 223 to 225 should be improved, and the first word of the second sentence in line 337 needs correction.

in Mathematical Models
The term "alpha" defined in Equation 2 might cause confusion as it is redefined in Equation 6.a. Consider using a different term for one of these occurrences.
The value of alpha is provided as 0.4, but the justification for this value appears later in the paper. A forward reference to the relevant section would help maintain continuity.

in Method for Detecting Similarity in Long Texts
Terms "N" and "M" are used in line 344 but defined only later. Introducing these terms earlier in the text would improve comprehension.

in Experiment
In Table 5, there is no need to include "TPU" in the header, as it is not used in the experiments.

in Discussion
Authors should correct the phrase "longe-sized texts" at the end of line 501.

References
Many references appear incomplete, such as those on lines 593, 604, 605, 613, 618, 620, 622, 626, 627, and others up to line 732. These should be revised to ensure they are complete and formatted correctly.

Experimental design

The research question is well defined and relevant. Long texts semantic similarity has received almost no attention as mentioned in the manuscript. The propose approach is novel. However, some points still need clarifications:
How were the 72 documents selected? Were specific criteria applied, or was the selection random?
On line 408, why is alpha set to 0.3 instead of 0.4? Authors should clarify this choice.
Why were two different setups used for the experiments? The second setup appears sufficient for all experiments; using a single setup would streamline the methodology.
In the creation of the gold standard (from line 776), since the process is known to be time-consuming, authors mentioned the fact that they used commonsense. Please, provide more information.

Validity of the findings

The authors have provided all the necessary information for replicating the experiment. The criteria for evaluating the accuracy of the results are diverse and well-defined. The results of the ablation study are thoroughly reported and appear plausible. Additionally, the authors have acknowledged the limitations of their work, particularly the reliance on the training model.

Cite this review as

Reviewer 2 ·

Basic reporting

The paper proposes a method to measure semantic similarity between long texts using sentence transformers and fuzzy logic to handle noise and computational constraints. Texts are split into sentences or chunks, and pairwise cosine similarity is calculated between their embeddings. Irrelevant pairs are filtered out based on a noise threshold, while relevant pairs are classified into four categories (non-related, concept-related, same topic, and identical) using fuzzy membership functions. The method is clearly explained. I have some remarks to take into account:
The method’s first step mentions segmenting the text into “sentences or chunks” based on contained words. Later, however, the algorithm always works on sentences. Is the method is designed to operate on both fixed-size chunks and natural sentences? This seems to be inconsistent, the pseudocode should reflect the two alternatives or clearly explain when one is used over the other.
In the algorithm and pseudocode, the complexity is discussed in terms of the number of sentences (L₁ × L₂). Later, in Section 5.2, the complexity is stated as O(N×M), where N and M are the numbers of words in each document. What is the actual unit of comparison and the associated computational cost?
The method requires fuzzy parameters for classifying similarity (e.g., non-related, concept-related, same topic, and identical). In the body text the parameters are noted as “[a], [b,b ], [c,g ], [d ],” but in the pseudocode (line 22) the function call introduces additional parameters (alpha, beta, gamma, delta) without clear explanation. A clarification of parameters that are needed in each step is required.
Some typos:
Citation punctuation and style vary (for example, some author lists use semicolons while others do not).
In Section 3, a phrase such as “a small language can be exploited” is likely intended to refer to a “small language model”.
The description “documents with sizes up to 105” is unclear. Maybe a unit is forgotten.
The pseudocode initializes an array as PairData.Deciles (with uppercase “D”) but later refers to it as PairData.deciles (lowercase “d”). Consistency here is important for implementation.

Experimental design

In relation to the previous remark about parameters, the tuning process for the fuzzy logic parameters ((alpha, beta, gamma, delta) is vague. Empirical tuning is described, but the lack of detailed methodology leaves the process unclear and non-replicable. The method relies heavily on these parameters for classification but does not provide a strong justification or systematic process for setting them.
The paper does not incorporate standard evaluation metrics (e.g., precision, recall, F1-score) to quantify the performance of the method. Instead, it focuses on fuzzy membership degrees, which are not commonly used for benchmarking. This limits the ability to compare clearly the method against other approaches. The authors should compare based on standard metrics.
Chunk sizes for large models are arbitrarily chosen. This may potentially disadvantaging them and skewing comparisons. The authors should justify the selection and conduct sensitivity analysis to assess its impact on the results.
The threshold for filtering noise (alpha = 0.4) is set empirically without justification. What does “empirically” mean? The paper does not provide a detailed analysis of why this value was chosen or its sensitivity.

Validity of the findings

The creation of the gold standard relies on "commonsense" labeling, which introduces subjectivity and potential bias. Since the method emphasizes precision in similarity measurement, the authors should comment on the subjective labeling of the dataset that may undermines the reliability of the evaluation.
The method is described as computationally efficient, but no detailed runtime analysis or benchmarks are provided to support this claim. The authors should show some runtime comparison and resource usage in the the results.
The comparison between small models and large models is unfair due to arbitrary chunk size selection and potential training biases or suboptimal hyperparameters for larger models. The claim that smaller models "outperform" larger ones may not hold universally without a more balanced setup.

Cite this review as

---

## Round 0.2 · Minor Revisions

Reviewer 2 has become satisfied with the required changes in the text; however, s/he proposes to improve several (minor) issues and I completely support their opinion. I am sure that the authors will accept the suggestions to improve their manuscript even more.

Reviewer 2 ·

Basic reporting

• The manuscript uses clear, professional English and provides sufficient background with well-cited, relevant literature.
• The structure is professional, with appropriate figures, tables, and shared publically raw data and code for reproducibility.
• Key terms and formulas are clearly defined: no formal theorems are presented, the theoretical components are appropriately explained.

Experimental design

• The manuscript presents original research aligned with the journal scope and addresses a clear and relevant research question on long-text semantic similarity.
• It identifies a meaningful gap, limited work on long-text similarity, and proposes a novel fuzzy logic-based method to address it.
• The investigation is technically sound, and the methods are described in enough detail to allow replication, with supplemental materials supporting implementation.

The authors have made commendable efforts in addressing most of my comments. The revised manuscript significantly improves in terms of clarity, consistency, and evaluation rigor. Notably, the authors clarified the use of sentence-based preprocessing over chunks, provided a detailed breakdown of model performance, and incorporated standard evaluation metrics (precision, recall, F1-score, accuracy) across similarity classes. They also improved the consistency of pseudocode and corrected typographical and citation issues. Furthermore, the response letter clearly outlines where extended material was added to the supplemental files, helping mitigate length constraints. However, a few important issues remain partially addressed or unresolved:

• Threshold parameter justification (Q26): The “empirical” selection of the alpha threshold is now justified by text. The authors could present supporting visualizations (e.g., similarity distribution plots) or a sensitivity analysis to demonstrate how alpha impacts classification.
• Chunking vs sentence-based preprocessing (Q16): The response explains the rationale for using sentences, but the method section and pseudocode still leave ambiguity about whether chunking is an option.

Validity of the findings

• The manuscript provides a clear rationale for replication, emphasizes the need for methods addressing long-text similarity and offers openly accessible data and code.
• All underlying data are robust, well-documented, and statistically sound, supporting the validity of the findings.
• Conclusions are logically derived, address the research question to a large extent, and are appropriately limited to the presented results.

• Evaluation metrics interpretation (Q24): Standard metrics are now reported, but the interpretation is insufficient. Some classes such as concept-related show very low F1-scores, and this performance gap is not discussed. The authors should reflect on these results, explain potential causes, and mention paths for improvement.
• Comparisons with large models (Q29): The authors acknowledge the limitations of their method with large language models (LLMs), but the performance comparison still appears unbalanced. To strengthen these claims, it should either attempt fairer chunk-level evaluations for LLMs, or reframe conclusions to emphasize that the current method is optimized for sentence transformers rather than asserting superiority over LLMs.

Cite this review as

---

## Round 0.3 · accepted · Accept

I am glad to see that careful following the recommendations of the reviewers contributed to improvement of the manuscript.

Reviewer 2 ·

Basic reporting

The manuscript is written in clear, professional English and follows a well-structured academic format. It shares the used data. It provides sufficient background and literature references to contextualize the study. All key terms and formulas are clearly defined, and the results are presented in a self-contained manner.

Experimental design

The manuscript presents original research that aligns well with the journal’s aims and scope and addresses a clearly defined and meaningful research question in the underexplored area of long-text semantic similarity as described in my previous review.

Validity of the findings

• The manuscript provides a clear rationale for replication, emphasizing the need for methods addressing long-text similarity and offering openly accessible data and code.
• All underlying data are robust, well-documented, and statistically sound, supporting the validity of the findings.
• Conclusions are logically derived, directly address the research question, and are appropriately limited to the presented results.

I have only one recommendation:
Low F1-scores for “concept-related” and “same topic” classes are not discussed. I recommend to explain why these scores are low (e.g., fuzzy threshold overlaps, lack of fine-tuning) and suggest improvements.

Cite this review as